# Advances in Pharmacological Properties, Molecular Mechanisms, and Bioavailability Strategies of Chlorogenic Acid in Cardiovascular Diseases Therapy

**DOI:** 10.3390/ph18091357

**Published:** 2025-09-11

**Authors:** Kai Huang, Duosu Zhang, Ruting Wang, Jiahao Duan, Long Hu, Fan Huang, Wei Liu, Jia Gu, Songlin Li, Chun Yang, Ling Yang

**Affiliations:** 1Department of Cardiology, The Third Affiliated Hospital of Soochow University, Changzhou 213003, China; 2Department of Cardiology, Changzhou Medical Center, Nanjing Medical University, Changzhou 213003, China; 3Department of Anesthesiology and Perioperative Medicine, The First Affiliated Hospital of Nanjing Medical University, Nanjing 210029, China

**Keywords:** chlorogenic acid, cardiovascular diseases, pharmacological action, cardiovascular activity, bioavailability

## Abstract

Cardiovascular diseases (CVDs), a group of global diseases, are characterized by high morbidity and mortality, imposing a significant burden on clinical practice. Chlorogenic acid (CGA), a natural compound composed of caffeic acid and quinic acid, is widely found in and extracted from plants such as *Lonicera japonica* (honeysuckle), *Eucommia ulmoides* (hardy rubber tree), tea leaves, and coffee beans. In recent years, increasing attention has been directed towards the pharmacological mechanisms of CGA in the treatment of CVDs. This review comprehensively summarizes the current knowledge on the preparation, metabolic pathways, pharmacological effects, and safety profile of CGA. Furthermore, it systematically analyzes the biological effects and molecular targets of CGA in the cardiovascular therapy and highlights strategies to enhance its bioavailability. These insights aim to provide a scientific basis for future basic research and clinical applications.

## 1. Introduction

Cardiovascular diseases (CVDs) are among the leading causes of death and disability worldwide, placing a significant burden on global healthcare systems and economies [1]. According to the Global Burden of Disease Study, as of 2019, the global number of CVDs cases reached 523 million (95% uncertainty interval [UI]: 497 to 550 million), and the loss of life due to death and disability result from CVDs has increased remarkably [2]. In the same year, more than 4 million deaths caused by CVDs were reported in China alone, the highest number globally [2,3]. Given the increasing prevalence and burden of CVDs, there is an urgent need to explore new and effective strategies for diagnosis, treatment, and prognosis.

Chlorogenic acid (CGA) is generally known as 5-caffeoylquinic acid (5-CQA), and CGAs are a class of natural compounds formed by the esterification of caffeic acid (CA) and quinic acid (QA). They are abundant in various plants, including *Lonicera japonica*, *Eucommia ulmoides*, tea leaves, and coffee beans [4]. Beyond the primary form, 5-CQA, studies have revealed that CGAs also include other derivatives, such as feruloylquinic acids (FQAs) and dicaffeoylquinic acids (diCQAs) [5]. Moreover, CGA exists in multiple isomeric forms, including neochlorogenic acid (nCGA, 3-CQA) and cryptochlorogenic acid (cCGA, 4-CQA) [6] (Figure 1). As a natural compound with extensive distribution, CGA exhibits potent anti-inflammatory and antioxidant activities [7], making it a promising candidate for CVDs research. With continuous progress in pharmacological studies, a comprehensive analysis of CGA’s recent advances in cardiovascular research is warranted.

This review first outlines the preparation methods and metabolic pathways of CGA. Next, it discusses the cardiovascular pharmacological activities of CGA and elucidates its underlying mechanisms to assess its therapeutic potential in CVDs management. Additionally, we retrospect the safety profile of CGA to highlight its current limitations in clinical practice. Finally, we summarize the strategies to enhance the bioavailability of CGA to facilitate its clinical translation. In conclusion, this review aims to provide new insights into the application of CGA in CVDs based on recent advances in research.

A systematic literature search was conducted using the keywords “Chlorogenic acid”, “Cardiovascular diseases”, “Chlorogenic acid”, “Pharmacological action”, “Chlorogenic acid”, “Cardiovascular activity”, “Chlorogenic acid”, and “Bioavailability” in the PubMed and Web of Science databases, covering the period from January 2000 to June 2025. The inclusion criteria were as follows: (1) studies investigating chlorogenic acid and its derivatives in relation to cardiovascular diseases; (2) publications with robust methodology, conclusive findings, and high thematic relevance. The exclusion criteria were studies with major methodological flaws or dubious credibility. Ultimately, 95 publications met the criteria for inclusion.

## 2. Preparation and Metabolic Pathways of CGA

### 2.1. Preparation

As a naturally occurring compound found in plants, efficient extraction of CGA is crucial. In recent years, besides the traditional heat-reflux extraction method, several novel preparation techniques have been developed, including microwave-assisted extraction, infrared-assisted extraction, ultrasonic-assisted extraction, counter-current chromatography, and centrifugal partition chromatography, etc. [8]. Currently, natural extraction methods remain the primary means of obtaining CGA, as the compound’s complex biosynthetic pathway and chemical instability limit the artificial chemical synthesis. It is necessary to overcome these limitations in further technological advancements. Thus, continuous improvement in preparation techniques will contribute to unlocking the potential value of CGA in healthcare and pharmaceutical industries.

### 2.2. Metabolic Pathways

The absorption and metabolism of CGA are fundamental to its beneficial pharmacological effects. Similarly to many other natural compounds, CGA follows complex metabolic pathways. As a dietary component, CGA exhibits moderate oral bioavailability [9], and its molecular structure remains largely intact during oral digestion [10]. In the gastrointestinal tract, CGA is absorbed into the bloodstream through multiple mechanisms: approximately one-third of ingested CGA is absorbed intact in the upper digestive tract, while the remaining fraction undergoes partial hydrolysis and absorption in the lower gastrointestinal tract before entering systemic circulation and being further metabolized in the liver [8]. The composition of gut microbiota plays a critical role in CGA’s metabolism and absorption, producing 11 key metabolites, with the most primary products dihydrocaffeic acid, dihydroferulic acid, and 3-(3-hydroxyphenyl) propionic acid [11]. In plasma, the major metabolites of CGA include CA and FA. These metabolites serve as critical mediators underlying the pharmacological benefits of CGA [12]. A clinical study revealed that, following the ingestion of 269.5 mg of CGAs through coffee, as many as 25 metabolites were identified in plasma and 42 in urine, primarily in the forms of sulfate and methyl derivatives [13]. Meanwhile, human studies have shown that age, sex, health status, and genetics factors contribute to an inter-individual variability in the metabolism of polyphenols. Therefore, it is imperative that future research on metabolism of CGAs adopts an integrative approach which adequately accounts for inter-individual variability.

## 3. Cardiovascular Activity of CGA

Since the first study reported the antihypertensive effects of CGA in spontaneously hypertensive rats (SHRs) in 2002 [14], increasing attention has been directed towards CGA’s role in cardiovascular research. Recent pharmacological investigations have uncovered CGA’s unique cardiovascular activity, including antihypertensive effects, regulation of atherosclerosis, prevention of myocardial infarction, and improvement of heart failure outcomes [15].

### 3.1. Hypertensive

Hypertension is one of the most common CVDs, with a rising prevalence, especially in low- and middle-income countries compared with high-income nations [16]. Considerable studies have demonstrated the antihypertensive potential of CGA. For instance, CGA has been shown to effectively reduce blood pressure of mice with high-fructose-induced, salt-sensitive hypertension [17]. A meta-analysis further confirmed that CGA supplementation has a statistically significant antihypertensive effect in humans [18]. It was reported that CGA lowers blood pressure by relaxing vascular smooth muscle and improving endothelial function, demonstrating a significant effect on hypertension management [19]. CGA has also been shown to ameliorate hypertension induced by intraperitoneal injection of cyclosporine in mice [20], likely by reducing the activity of angiotensin-converting enzyme and limiting the production of angiotensin II consequently. Additionally, CGA enhances the activity of endothelial nitric oxide synthase by reducing cholinesterase and arginase activity, which helps to restore nitric oxide (NO) bioavailability suppressed by cyclosporine. In SHRs, Dietary 5-CQA has been shown to enhance NO bioavailability by inhibiting the sources of reactive oxygen species (ROS), such as NADPH oxidase, and the production of superoxide anions to prevent the decomposition of NO into peroxynitrite [21]. As NO is essential for vascular relaxation, it activates guanylate cyclase in vascular smooth muscle cells, raising intracellular cyclic guanosine monophosphate (cGMP) levels. This increase in cGMP triggers a signaling cascade that lowers intracellular calcium concentrations, resulting in smooth muscle relaxation and vasodilation [22]. Zhu et al. [17] demonstrated that the antihypertensive effects of CGA are significantly associated with the regulation of gut microbiota (specifically the enrichment of *Klebsiella*) and their bile acid metabolites. Although the exact mechanism is not fully understood, it may be related to the inflammatory state triggered by the dysfunction of gut microbiota metabolism [23].

However, not all studies have yielded positive results. For example, although coffees, which contained CGAs, enhanced plasma antioxidant capacity in healthy subjects with normal blood pressure, they did not achieve a significant antihypertensive effect, suggesting that CGAs may exert its blood pressure-lowering action primarily in hypertensive subjects [24]. Contradictorily, a placebo-controlled study showed that acute intake of coffee high in CGA and low in hydroxyhydroquinone improved the endothelial function in patients with stage 1 hypertension but did not significantly affect the systolic blood pressure [25]. This indicates that the antihypertensive effects of CGA might be a long-term outcome. Taken together, while CGA is promising as an antihypertensive agent, the underlying mechanisms should be fully elucidated.

### 3.2. Atherosclerosis

Atherosclerosis, a chronic inflammatory vascular lesion, is characterized by endothelial injury, lipid accumulation, and vascular calcification [26]. Atherosclerotic plaque narrows arterial lumens and stiffens vessel walls, eventually leading to ischemic diseases. A trend analysis from 1990 to 2019 indicates a growing global burden of atherosclerosis, highlighting the need for early prevention and therapeutic intervention to slower disease progression [27]. CGA has anti-inflammatory, antioxidant, antiplatelet, and lipid-lowering activities, which contribute to its vascular protective effects and its ability to prevent atherosclerosis development [28]. This is the reason why CGA is widely regarded as a potential natural anti-atherosclerotic agent. Its mechanisms of action may involve inhibiting the oxidation of low-density lipoprotein (LDL) and reducing endothelial damage caused by oxidized LDL (Ox-LDL). CGA also mitigates leukocyte migration, promotes cholesterol efflux from macrophages, and suppresses platelet aggregation and platelet–leukocyte interactions [15]. Studies have shown that CGAs, the major antioxidant in coffee, may be associated with reduced levels of plasma oxysterols (a group of biomarkers linked to atherosclerosis progression) and free fatty acids, which are known risk factors for CVDs [29]. In the apolipoprotein E knockout mouse model of atherosclerosis, CGA at 400 mg/kg significantly reduced serum levels of total cholesterol, triglycerides, and LDL-cholesterol, as well as inflammatory factors, thereby attenuating the aortic root dilation and reducing the severity of atherosclerotic lesion. The therapeutic effect was comparable to that of atorvastatin. Furthermore, CGA at 10 μM was found to prevent the formation of foam cells induced by Ox-LDL in RAW264.7 macrophages and promoted cholesterol efflux [30]. Additionally, Ca^2+^, a key mediator of platelet aggregation, the upregulation of its antagonists, such as cyclic adenosine monophosphate (cAMP) and cGMP, and the downregulation of thromboxane A2 plays a crucial role in CGA-mediated inhibition of platelet aggregation. This mechanism also inhibits platelet–leukocyte interactions [31]. Furthermore, a randomized placebo-controlled trial demonstrated that, compared to the control group, the subjects who consumed a CGA-containing green coffee bean extract for two weeks showed significant improvement in arterial stiffness by measuring the cardio-ankle vascular index [32].

In summary, CGA exerts anti-atherosclerotic effects by protecting endothelium, reducing vascular lipid deposition, preventing thrombus formation, and improving arterial stiffness.

### 3.3. Ischemia/Reperfusion Injury

Ischemia/reperfusion (I/R) injury is a common complication of coronary artery revascularization following acute occlusion, with a complex pathogenesis closely associated with inflammation [33]. In line with previous studies, the production of inflammatory factors is a major contributor to I/R injury, and the inflammatory response is throughout the whole pathological process [34]. Given CGA’s profound anti-inflammatory and antioxidant properties, it holds great potential for mitigating I/R injury. Existing studies have demonstrated that CGA alleviates I/R injury in multiple organs, including the liver [35], brain [36], and kidneys [37]. On the side, CGA has been found to prevent myocardial I/R injury both in vivo and in vitro by suppressing the expression of the long non-coding RNA Neat1 (Lnc Neat1) and the NOD-like receptor (NLR) with a pyrin domain 3 (NLRP3) inflammasome-mediated pyroptosis pathway. Notably, Lnc Neat1 expression is upregulated in myocardial tissue during I/R injury, promoting cardiomyocyte apoptosis and triggering NLRP3 inflammasome activation, thereby amplifying the inflammatory cascade and then exacerbating tissue damage [38]. In addition, CGA, nCGA, and cCGA demonstrated protective effects against cardiac I/R injury in vitro DPPH (2,2-diphenyl-1-picrylhydrazyl) assay. Among them, cCGA exhibited the strongest antioxidant effect, which may be related to improved mitochondrial function [39].

Despite these promising findings, research on CGA’s role in myocardial I/R injury remains limited. Future studies should focus on elucidating the precise molecular mechanisms underlying CGA’s cardioprotective effects to provide a stronger foundation for clinical application.

### 3.4. Myocardial Infarction

Myocardial infarction (MI) results from myocardial necrosis due to ischemia and hypoxia caused by coronary artery stenosis or occlusion. Acute myocardial infarction (AMI) is the most common clinical condition [40]. Since MI causes irreversible damage, effective prevention strategies have received extensive attention. Currently, CGA has shown promising results in preventing MI. Akila et al. [41] revealed that CGA pretreatment exerts protective effects in the isoproterenol (ISO)-induced MI rat model by reducing oxidative stress in myocardial cell membranes, mitochondrial membranes, and lysosomal membranes. This protection is achieved by decreasing ROS and lipid peroxidation. Another research showed ISO-induced MI in rats is associated with reduced activity of antioxidant enzymes such as superoxide dismutase (SOD), catalase (CAT), glutathione peroxidase, and glutathione-S-transferase, as well as a decline in non-enzymatic antioxidants like vitamins C and E and glutathione (GSH), then this imbalance leads to the accumulation of ROS and various lipid peroxidation, aggravating myocardial damage. Finally, TTC (2, 3, 5-triphenyltetrazolium chloride) staining and histopathological results confirmed that pretreatment with CGA ameliorates tissue damage and significantly reduces myocardial infarct size [42]. Furthermore, CGA has been shown to reduce infarct size in MI induced by left anterior descending artery (LAD) ligation in rats. It achieves this by suppressing inflammation and enhancing the activity of antioxidant enzymes, such as SOD and CAT, thereby improving cardiac function [43]. In addition, CGA has been observed to enhance myocardial contractility and reduce fibrosis in MI mice, improving ventricular remodeling. This protective effect may be attributed to reduced macrophage infiltration and inflammatory damage within myocardial tissue without a clear dose dependency [44].

Generally speaking, CGA’s anti-inflammatory and antioxidant properties play a crucial role in the prevention of MI, providing a solid scientific basis for its potential application as a preventive strategy for MI in the future.

### 3.5. Heart Failure

Heart failure (HF) is the terminal stage of many cardiovascular diseases and poses a severe threat to human health as a complex clinical syndrome [45]. Myocardial hypertrophy is an early stage of HF, and preventing hypertrophy is essential for halting HF progression. CGA has been identified as a potential candidate for preventing HF by inhibiting myocardial hypertrophy [46]. It was demonstrated that CGA alleviated ISO-induced cardiomyocyte hypertrophy by suppressing endoplasmic reticulum stress [47]. Studies indicate that oxidative stress induced by excessive ROS generation could lead to cardiomyocyte damage and apoptosis, thereby promoting HF development [48]. It is necessary to emphasize CGA mitigates ISO-induced hypertrophy by scavenging ROS and ameliorating inflammatory responses [49]. Furthermore, CGA has been shown to significantly improve cardiac function and reduce hypertrophy and fibrosis in a transverse aortic constriction (TAC)-induced HF mouse model by attenuating ferroptosis [50]. Ferroptosis, a novel form of regulated cell death, is associated with abnormalities in iron, lipid, and mitochondrial metabolism, along with excessive ROS accumulation. It results in cellular death due to impaired redox homeostasis, primarily through GSH depletion. Studies have confirmed that ferroptosis plays a crucial role in HF development [51]. CGA’s ability to alleviate ferroptosis provides new insights and therapeutic targets for HF management.

In conclusion, noteworthy advances in CGA research over recent years have laid a solid foundation for its clinical application. While CGA exhibits pharmacological effects across a range of cardiovascular and other systemic diseases, its antioxidant and anti-inflammatory properties underpin these benefits (Figure 2). Therefore, future research should focus on unraveling the deeper molecular mechanisms of CGA’s pharmacological actions.

## 4. Regulation of Cardiovascular Signaling Pathways by CGA

According to current studies, CGA exerts significant cardioprotective effects by modulating multiple signaling pathways. It has been reported that CGA could alleviate tumor necrosis factor-α (TNF-α) -induced cardiomyocyte injury by inhibiting the NF-κB and JNK signaling pathways, highlighting its cardioprotective potential in a TAC mouse model [52]. Zhang et al. [53] found that activation of the NF-κB and JNK pathways upregulates the expression of pro-inflammatory cytokines, such as TNF-α and IL-6, and then prolongs the inflammatory response, leading to increased myocardial necrosis thereby exacerbating MI eventually. CQA derivatives have been shown to significantly ameliorate these effects by regulating the NF-κB and JNK pathways. Moreover, CGA binds to adenosine A2A receptor on the platelet and activates AC/cAMP/PKA signaling pathway to exert antiplatelet effect, thus having the potential to improve atherosclerosis and prevent MI [54]. In another study, He et al. [46] demonstrated that pretreatment of AC16 cells with CGA alleviated ISO-induced hypertrophy by inhibiting the Wnt/β-catenin signaling pathway to downregulate the expression of β-catenin and other core proteins. Similarly, CGA, cCGA, nCGA and 3,5-di-CQA which are the effective constituents from Si-Miao-Yong-An decoction have been shown to suppress the Akt/mTOR/HIF-1α signaling pathway, reducing ISO-induced hypertrophy and decreasing atrial natriuretic peptide levels, suggesting a potential role in heart failure prevention [55]. Interestingly, cCGA and its active metabolites, CA and 4-FQA, also exhibited cardioprotective effects by inhibiting the Akt/mTOR/HIF-1α pathway [56]. Furthermore, 3,5-diCQA was found to prevent tert-butyl hydroperoxide-induced apoptosis in H9C2 cardiomyocytes by upregulating the PI3K/Akt pathway, thereby reducing tissue damage in a dose-dependent manner [57]. Activation of the PI3K/Akt pathway enhances the expression of anti-apoptotic protein Bcl-2 and downregulates the expression of pro-apoptotic protein Bax, providing protection against apoptosis [58]. Furthermore, CGA reduces oxidative stress and ROS-induced damage by upregulating the Nrf2/HO-1 pathway, thereby mitigating doxorubicin-induced cardiotoxicity and improving cardiac tissue integrity [59].

Future research should explore the interactions between these pathways (Figure 3) and the specific regulatory mechanisms of CGA across different cardiovascular disease models.

## 5. Safety Research

Since CGA is found in traditional Chinese medicinal plants such as *Lonicera japonica* and *Eucommia ulmoides*, which are widely used in traditional Chinese medicine [60,61], it is generally considered safe. As a natural compound with various pharmacological effects, most preclinical and clinical studies have reported no significant adverse reactions to CGA [62]. For example, CGAs from coffee beverage, known for their potent lipid-lowering effect, have shown high oral safety at low doses (407 mg/day) while preventing atherosclerosis [63]. However, toxicological studies on CGA in humans remain insufficient, as clinical studies tend to focus more on its beneficial activities rather than toxicity. In a toxicity study using standardized decaffeinated green coffee bean extract (containing 50% CGAs), no toxicity was observed in acute oral toxicity tests with a single dose of 2 g/kg in rats. In a 90-day sub-chronic toxicity evaluation, CGA at doses of 0.25, 0.5, and 1 g/kg caused changes in food intake, brain and spleen weights, and hematological and biochemical parameters, but these remained within physiological ranges without toxic responses [64]. Although CGA has not exhibited significant toxic effects at dietary or low doses, acute toxicity studies have reported a median lethal dose (LD_50_) of >2000 mg/kg in rats, with similar results in mice, indicating potential toxicity at high doses [65]. Additionally, in humans, high doses of CGA (2 g/day) were found to increase plasma homocysteine levels, a known risk factor for cardiovascular diseases [66]. An in vitro study demonstrated that CGA induced endothelium-dependent vasodilation in the rat aortic ring, contributing to its positive antihypertensive effects. However, paradoxically, high concentrations of CGA (10^−4^ mol/L) impaired acetylcholine- and bradykinin-induced vasodilation, indicating potential cardiovascular toxicity at elevated concentrations [67].

Given these findings, in clinical practice, caution is advised regarding the use of high-dose CGAs due to the potential risk of drug interactions, particularly with anticoagulant or antiplatelet medications which may increase the risk of bleeding and antihypertensive agents which may potentiate the risk of hypotension. Future clinical studies should carefully monitor the use of high-dose CGA and establish safe dosage ranges to provide more precise guidance for clinical applications.

## 6. Strategies to Improve the Bioavailability of CGA

Although CGA holds great potential for the treatment of CVDs, several intrinsic limitations, including molecular instability, variable chemical properties, suboptimal lipid solubility, and low bioavailability, hinder its clinical applications [68]. Essentially, as a hydrophilic molecule, CGA struggles to cross lipophilic membranes, resulting in poor absorption and bioavailability [69]. Simply increasing the oral dose is not an advisable solution, as it carries significant risks. Therefore, structural modification and the optimization of drug delivery systems are urgent tasks to improve CGA’s bioavailability.

### 6.1. Structural Modification

While CGA possesses several isomers and derivatives with distinct pharmacological effects, it is still arduous to overcome the structural limitations that affect its bioavailability. Developing novel and rational structures for CGA is essential to enhance its bioavailability. Recent studies have revealed the feasibility of improving CGA’s bioavailability through structural modification [70]. According to Fu et al. [71], CGA-gelatin conjugates, formed via dehydration condensation, exhibited higher antioxidant activity than free CGA while reserving CGA’s antibacterial properties. Similarly, an in vitro study reported that the covalent bonding between CGA and soluble oat β-glucan significantly improved CGA’s structural stability and maximized its pharmacological potential [72]. In addition, CGA has been found to form a chlorogenic acid–phospholipid complex (CGA-PC) through hydrogen bonding and hydrophobic interactions, which exhibited higher oral bioavailability than free CGA. CGA-PC also mitigated mitochondrial ROS accumulation and respiratory deficits in SAMP8 mice with I/R injury, reducing inflammatory cascades and providing strong cardioprotection for aging hearts [73].

Modifying CGA’s structure improves its stability and lipid solubility, significantly enhancing its bioavailability and clinical potential.

### 6.2. Optimization of Drug Delivery Systems

In addition to structural modifications, researchers have explored various encapsulation and delivery systems for CGA, including liposomes [74], hydrogels [75], and nanoparticles [76], to enhance bioavailability.

#### 6.2.1. Liposomes

Liposomes, composed of phospholipid bilayer and similar to the cell membrane, exhibit both hydrophilic and lipophilic properties, making them ideal delivery carriers for drugs [77]. Liposomes offer high biocompatibility and biodegradability, making them a widely used platform for drug delivery systems. They can encapsulate multiple active compounds, facilitate rapid absorption, extend intracellular retention, and improve bioavailability while minimizing cytotoxicity [78]. Studies have reported that CGA-loaded liposomes, prepared from cholesterol and phosphatidylcholine, showed a relative oral bioavailability of 129.38% compared to free CGA. These liposomes also significantly enhanced antioxidant activity in a CCl_4_-induced hepatotoxicity model in mice [79]. Furthermore, liposomal encapsulation has been shown to enhance CGA’s anti-inflammatory [80] and anticancer properties [81]. Liposomes, with their ability to improve lipid solubility and ease of preparation, hold great potential for future applications.

#### 6.2.2. Lipid Microspheres

Lipid microspheres, derived from intravenous nutrition emulsions, are single-layer molecular dispersions with a median diameter of about 200 nm. They consist of a phospholipid-coated fatty oil core and serve as an efficient drug delivery system [82]. Lipid microspheres have been utilized in various clinical drugs like alprostadil [83] and flurbiprofen ester [84], offering significant delivery advantages. Drugs are dissolved in the oil phase or at the interface of lipid microspheres, which could avoid vascular irritation, accumulate in rough vascular walls, tumors, or inflammatory sites, and easily cross membranes to control drug release [85]. These properties make lipid microspheres not only sustainable and efficient, but also improve the safety of medication, which can reduce the drug dose and the occurrence of adverse reactions. Although no studies have yet reported the use of lipid microspheres for CGA delivery, their successful applications in other drug fields suggest that they could become effective carriers for CGA in the future.

#### 6.2.3. Self-Emulsifying Drug Delivery System

Self-emulsifying drug delivery system (SEDDS) is a kind of isotropic mixture of drugs, synthetic or natural oils, surfactants, and co-surfactants that spontaneously form nanoscale droplets in contact with gastrointestinal fluids [86]. SEDDS protects active ingredients from degradation in the gastrointestinal environment and enhances cellular uptake. Initially, SEDDS emerged as a method to significantly improve the oral bioavailability of hydrophobic actives, but it has recently gained attention for hydrophilic drugs through hydrophobic ion pairing [87]. Although some researchers distinguish between SEDDS and self-microemulsifying drug delivery system (SMEDDS) as translucent and transparent emulsions, both have been widely used interchangeably in numerous studies [88]. One study showed that SMEDDS significantly improved CGA’s oral bioavailability (249.4%) compared to the CGA suspension. Interestingly, the system altered CGA’s tissue distribution, enhancing renal targeting (relative uptake efficiency of 2.79) [89]. This finding suggested that SMEDDS improves CGA absorption and slows metabolism. Above all, SEDDS/SMEDDS is a promising strategy for oral CGA delivery.

#### 6.2.4. Hydrogels

Hydrogels are scaffold-like biomaterials composed of biological polymer networks that retain large amounts of water, and they are widely used in biomedicine for drug delivery [90]. Hydrogels possess versatile physical, chemical, and biological properties, allowing for precise control and sustained release of active compounds [91]. Typically constructed from polysaccharides or proteins, hydrogels protect active substances during delivery, enhancing bioavailability and improving delivery efficiency [92]. Recent studies have reported that CGA-loaded hydrogels exhibit excellent biocompatibility, self-healing properties, and significant anti-inflammatory, antioxidant, and antibacterial effects. These hydrogels promote macrophage polarization from pro-inflammatory M1 to anti-inflammatory M2 phenotypes, accelerating wound healing in rats [75,93]. While most studies have focused on topical applications through direct injection, hydrogels may represent a novel route for CGA delivery.

#### 6.2.5. Nanoparticles

Nanoparticles are ultra-fine particles ranging from 1 to 100 nm in size, characterized by the properties of environmentally friendly, cost-effective, and scalable manufacturing [94]. As protective carriers, nanoparticles prevent the degradation of active substances during digestion, improve cellular uptake, allow controlled release, and enable targeted delivery [92]. Roy et al. [94] demonstrated that bovine serum albumin (BSA)-decorated chlorogenic acid silver nanoparticles (AgNPs-CGA-BSA) exhibited significant antioxidant and anticancer effects both in vitro and in vivo. In addition to binding to Ag^2+^, calcium–chlorogenic acid nanoparticles (Ca-CGA NPs), synthesized through chelation, showed improved bioavailability and enhanced anti-inflammatory activity, accelerating bone repair [95]. Nanoparticle-based delivery systems offer a promising platform for CGA delivery in the field of CVDs.

In conclusion, the modification of CGA’s structure and optimization of its drug delivery systems significantly enhances its cardiovascular activity, unlocking its vast potential for applications in the field of CVDs.

## 7. Conclusions and Future Perspectives

In recent years, research on CGA in the domain of CVDs has deepened significantly. Numerous studies have demonstrated that CGA exerts various cardioprotective effects due to its potent anti-inflammatory and antioxidant properties, playing an active role in several CVDs. However, research on the specific signaling pathways in different CVDs remains incomplete, and further exploration is needed to identify other potential therapeutic targets in the CVDs domain.

Currently, clinical research data on CGA remain relatively limited (Appendix A), because most studies rely on animal models. Although no significant toxic effects have been observed so far, the safety of CGA for clinical applications still requires further evaluation. Moreover, the physicochemical properties and metabolic characteristics of CGA result in low bioavailability, which restricts its clinical utility. Future studies should focus on modifying drug structures and optimizing delivery systems to enhance CGA’s bioavailability and maximize its therapeutic potential. Given the gut microbiota’s importance, developing targeted interventions (e.g., prebiotics) could maximize the metabolic efficiency and health benefits of CGA. It is important to note that increasing bioavailability may enhance CGA’s pharmacological effects but could also reveal potential adverse effects. Therefore, determining the optimal dosage will be crucial for guiding future clinical studies. Concurrently, subtle structural variations among CGAs may lead to differential metabolic conversion, absorption profiles, cardiovascular bioactivities, and even toxicological implications. Nevertheless, comprehensive investigations about the distinct biological activities across CGAs are limited too. This knowledge gap necessitates deeper pharmacological characterization of CGAs.

Bottom line, CGA shows great potential for CVDs treatment, but further clinical research is needed to fully establish its therapeutic applications. With continued research, the pharmacological actions and mechanisms of CGA will become clearer, providing new insights and strategies for the prevention and treatment of CVDs.

## Figures and Tables

**Figure 1 pharmaceuticals-18-01357-f001:**
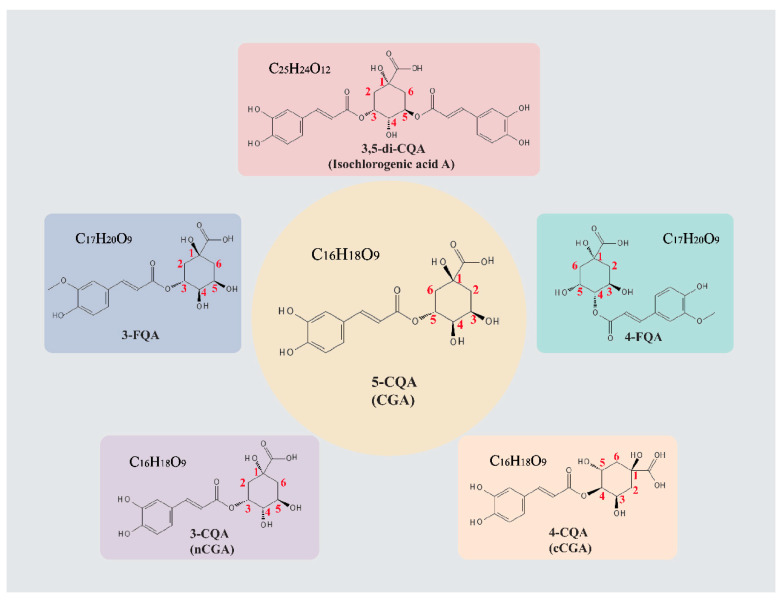
The chemical structure of CGAs. The chemical structures of CGA and its derivatives were obtained from PubChem (https://pubchem.ncbi.nlm.nih.gov/, accessed on 29 November 2024). The figures in red refer to the carbon number.

**Figure 2 pharmaceuticals-18-01357-f002:**
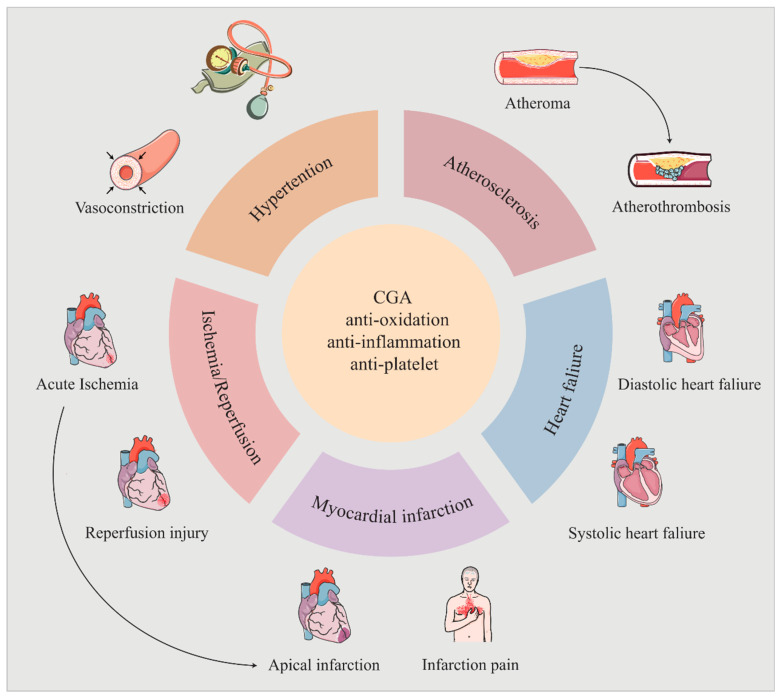
The cardioprotective effects of CGA. CGA exerts various pharmacological effects against CVDs through its anti-oxidation, anti-inflammation, and antiplatelet properties.

**Figure 3 pharmaceuticals-18-01357-f003:**
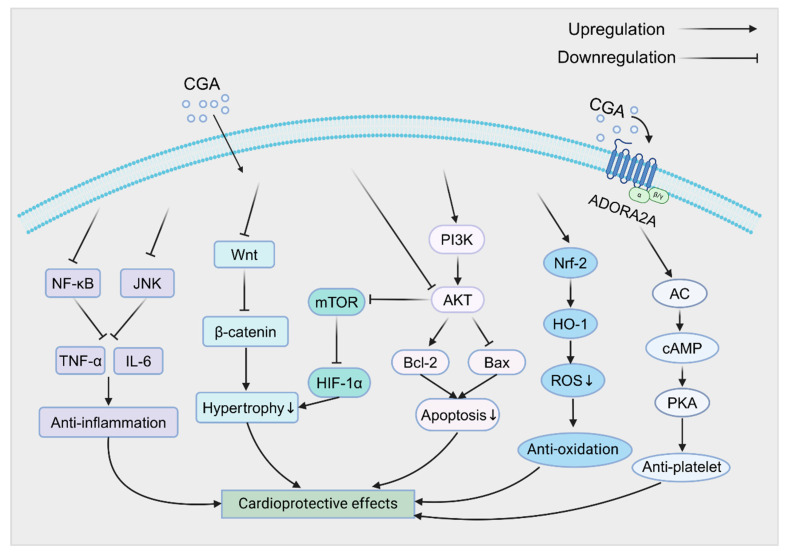
The cardioprotective mechanisms of CGA. CGA exhibits significant cardioprotective effects by modulating multiple signaling pathways. NF-κB, nuclear factor κB; JNK, c-Jun N-terminal kinase; TNF-α, tumor necrosis factor α; IL-6, interleukin-6; PI3K, phosphatidylinositol 3-kinase; Akt, protein kinase B; mTOR, mammalian target of rapamycin; HIF-1α, hypoxia-inducible factor 1α; Bcl-2, B-cell lymphoma-2; Bax, Bcl-2-associated X protein; Nrf-2, nuclear factor erythroid-2 related factor 2; HO-1, heme oxygenase-1; ROS, reactive oxygen species; ADORA2A, adenosine A2A receptor; AC, adenylate cyclase; cAMP, cyclic adenosine monophosphate; PKA, protein kinase A; ↓ means descending of the expression or effect. This figure was created with biorender (https://biorender.com/).

## Data Availability

No new data were created or analyzed in this study.

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
