# Peer review of "Advances in Pharmacological Properties, Molecular Mechanisms, and Bioavailability Strategies of Chlorogenic Acid in Cardiovascular Diseases Therapy"

_pharmaceuticals, 2025, doi:10.3390/ph18091357_

Round 1

Reviewer 1 Report

Comments and Suggestions for Authors

The authors summarized the potential of chlorogenic acids from plants for cardiovascular diseases. The manuscript is well-organized. However, there are still some points that can be considered for a better clarity and messages:

Major:

  1. Please keep in mind that chlorogenic acids is a chemical/compound class and not a single compound. The authors have mentioned the chemical characteristics of chlorogenic acids with several examples in the manuscript. But, in the next part the authors generalized CGA as single compound without further explanation whether it is 3-CQA, 5-CQA, etc. A slightly different in the chemical structure may lead to a different activity and it is important information that can be explored by the authors.
  2. The authors mentioned in section 2 that most of the preparation of the CGA are from extraction from natural sources. However, in the next parts, the authors only mentioned the CGA activity, mode of action, etc without giving information whether these information is obtained from a pure/single compound CGA or from an extract/marketed products. These additional information can be used to identify the natural sources of CGAs together with their potency. 
  3. Patients with CVDs often get a prescription of daily dosage of some drugs to improve their condition. The authors discussed the potential use of preparation with CGAs for the CVDs patient. Please add information on the possible interaction between CGAs with drugs in section 5.

Minor: 

  1. Please give atom number in figure 1 and replace the figure with the better quality for better readability.
  2. Please follow the instructions for authors in the writing of citation and references. 
  3. line 182 live -> liver

Author Response

The authors summarized the potential of chlorogenic acids from plants for cardiovascular diseases. The manuscript is well-organized. However, there are still some points that can be considered for a better clarity and messages:

We sincerely appreciate your comprehensive feedback on my manuscript. I will address each of your suggestions individually in my revisions. Additionally, the alterations according to the comments in revised manuscript have been highlighted or deleted and that the revision has been enclosed in the submission system.

Comment 1: Please keep in mind that chlorogenic acids is a chemical/compound class and not a single compound. The authors have mentioned the chemical characteristics of chlorogenic acids with several examples in the manuscript. But, in the next part the authors generalized CGA as single compound without further explanation whether it is 3-CQA, 5-CQA, etc. A slightly different in the chemical structure may lead to a different activity and it is important information that can be explored by the authors.

Response 1: We sincerely appreciate your comment. Chlorogenic acids is a chemical/compound class and not a single compound. We agree with this comment. However, as noted in our Introduction, CGA is generally known as 5-CQA unless otherwise specified, given that 5-CQA is the primary form of CGAs (this clarification has been incorporated into the section 1, and it can be found in the line 4 of the second paragraph).

It should be acknowledged that subtle structural variations among CGA isomers may lead to differential metabolic conversion, absorption, and cardiovascular bioactivity. For instance, as discussed in the article, cCGA exhibits the strongest antioxidant effect among CGA, nCGA and cCGA (line 15 of the section 3.3). Nevertheless, systematic comparisons of subtype-specific bioactivities remain scarce in the current literature, representing a recognized limitation in CGA research (this point has been added to Section 7 of the manuscript,it can be found in the line 13 of the second paragraph). Revisions have been incorporated throughout the manuscript to align with the formulations established in the references.

Comment 2: The authors mentioned in section 2 that most of the preparation of the CGA are from extraction from natural sources. However, in the next parts, the authors only mentioned the CGA activity, mode of action, etc without giving information whether these information is obtained from a pure/single compound CGA or from an extract/marketed products. These additional information can be used to identify the natural sources of CGAs together with their potency. 

Response 2: We sincerely appreciate your suggestion. CGA used in current research most derived from marketed products. Concurrently, multiple studies have utilized extracts isolated from natural sources or Chinese traditional herbal decoctions as test materials. Through comprehensive phytochemical analysis, revealing CGA as one of the primary bioactive constituents responsible for the observed effects. We have provided more detailed documentation of these sources in respective studies throughout the manuscript. Unless otherwise specified, the term CGA in this context refers to the high purified compound used in each referenced study. Revisions have been incorporated throughout the manuscript to align with the formulations established in the references.

Comment 3: Patients with CVDs often get a prescription of daily dosage of some drugs to improve their condition. The authors discussed the potential use of preparation with CGAs for the CVDs patient. Please add information on the possible interaction between CGAs with drugs in section 5.

Response 3: Thank you for your valuable feedback. We have now included a discussion addressing potential drug interactions between CGA and cardiovascular medications in Section 5 (such as anticoagulants and antihypertensive drugs). It can be found in the line 1 of the second paragraph.

Comment 4: Please give atom number in figure 1 and replace the figure with the better quality for better readability.

Response 4: Thank you for your suggestive comment. If I understand your comment correctly, you are suggesting that we provide the chemical formula of the compounds. We have now incorporated them into Figure 1.

Comment 5: Please follow the instructions for authors in the writing of citation and references. 

Response 5: We sincerely appreciate your suggestion, and have comprehensively addressed them throughout the document.

Comment 6: line 182 live -> liver

Response 6: Thank you for pointing this out. We have corrected.

Reviewer 2 Report

Comments and Suggestions for Authors

The authors of the paper “ Advances in Pharmacological Properties, Molecular Mechanisms, and Bioavailability Strategies of Chlorogenic Acid in Cardiovascular Diseases Therapy” review studies on the effect of Chlorogenic Acid in cardiovascular diseases. The topic is interesting but it is not original. Recent papers are already published results on Chlorogenic Acid and its potential effects (Compr Rev Food Sci Food Saf 2020 Nov;19(6):3130-3158; Nutrients 2024 Mar 23;16(7):924)

The authors should improve their paper.

-Define the criteria used in the review. Which data bases have been used? Describe inclusion criteria and exclusion criteria.

-The bioactive properties of Chlorogenic Acid described in the paper have been widely investigated using cells in culture and animal models. Figure 3 summarizes potential mechanisms of the cardioprotective mechanisms of CGA as suggested by in vitro studies. The authors should describe the limits of their paper. As pointed by the authors, In vivo studies have shown that bioavailability of Chlorogenic Acid is low. In addition  from the metabolism of Chlorogenic Acid the main metabolites caffeic acid and ferulic acid are produced. Interactions between Chlorogenic Acid and gut microbiota have been described.

-What is known about the biological properties of the main Chlorogenic Acid metabolites? Cardiovascular effects and protective mechanisms are confirmed at concentrations similar to those observed in human studies? what about metabolites produced by human gut microflora?

-Human studies have shown that age, sex, health status, genetics factors contribute to an inter-individual variability in the metabolism of polyphenols. All these aspects should be included and discussed in the article.

-The authors should also revise their paper and include data from clinical trials. The effects of Chlorogenic Acid have been already studied

Author Response

The authors of the paper “ Advances in Pharmacological Properties, Molecular Mechanisms, and Bioavailability Strategies of Chlorogenic Acid in Cardiovascular Diseases Therapy” review studies on the effect of Chlorogenic Acid in cardiovascular diseases. The topic is interesting but it is not original. Recent papers are already published results on Chlorogenic Acid and its potential effects (Compr Rev Food Sci Food Saf 2020 Nov;19(6):3130-3158; Nutrients 2024 Mar 23;16(7):924)

The authors should improve their paper.

We sincerely appreciate your comprehensive feedback on my manuscript. I will address each of your suggestions individually in my revisions. Additionally, the alterations according to the comments in revised manuscript have been highlighted or deleted and that the revision has been enclosed in the submission system.

Comment 1: Define the criteria used in the review. Which data bases have been used? Describe inclusion criteria and exclusion criteria.

Response 1: Thank you for your suggestive comment. We have added the inclusion criteria and exclusion criteria at the bottom of section 1. It is the paragraph 4.

Comment 2: The bioactive properties of Chlorogenic Acid described in the paper have been widely investigated using cells in culture and animal models. Figure 3 summarizes potential mechanisms of the cardioprotective mechanisms of CGA as suggested by in vitro studies. The authors should describe the limits of their paper. As pointed by the authors, In vivo studies have shown that bioavailability of Chlorogenic Acid is low. In addition from the metabolism of Chlorogenic Acid the main metabolites caffeic acid and ferulic acid are produced. Interactions between Chlorogenic Acid and gut microbiota have been described.

Response 2: Thank you for your suggestive comment. We have added extra limits of our review (It can be found in line 12 of paragraph 2 of section 7).

Comment 3: What is known about the biological properties of the main Chlorogenic Acid metabolites? Cardiovascular effects and protective mechanisms are confirmed at concentrations similar to those observed in human studies? what about metabolites produced by human gut microflora?

Response 3: Thank you for your valuable suggestion. Indeed, intestinal metabolites of CGA—such as ferulic acid, dihydrocaffeic acid, and dihydroferulic acid—exhibit multifaceted bioactivities including antioxidant, anti-inflammatory, free radical-scavenging, and mitochondrial protective effects. These compounds serve as critical mediators underlying the pharmacological benefits of CGA (It can be found in the line 13 of paragraph 1 of section 2.2).

Moreover, these protective mechanisms have been experimentally validated in both cellular and animal models at concentrations physiologically relevant to human exposure. However, it should be noted that current research on CGA remains largely dependent on preclinical studies, while clinical evidence is still relatively limited—a key constraint that we have acknowledged as a research limitation.

Additionally, gut microbiota-derived metabolites (e.g., hippurate and SCFAs) contribute to systemic regulation via the gut-vascular axis. We have also added content regarding the potential for developing microbiota-targeted interventional strategies—such as prebiotic formulations—to optimize the metabolic efficiency and functional benefits of CGA (It can be found in line 7 of paragraph 2 of section 7).

Comment 4: Human studies have shown that age, sex, health status, genetics factors contribute to an inter-individual variability in the metabolism of polyphenols. All these aspects should be included and discussed in the article.

Response 4: Thank you very much for your reminder. Considering the individual differences, we have added the corresponding discussion in the relevant part of the review (It can be found in line 17 of paragraph 1 of section 2.2).

Comment 5: The authors should also revise their paper and include data from clinical trials. The effects of Chlorogenic Acid have been already studied

Response 5: Thank you very much for your suggestion. Currently, clinical research data on CGA remain relatively limited. This review reported and analyzed a small number of clinical studies, and the summary table will be uploaded to the supplementary material (Table S1: Summary of clinical trials related to CGA.).

Round 2

Reviewer 1 Report

Comments and Suggestions for Authors

The authors have revised the manuscript based on the comments from the reviewer. There are only several minor changes needed:

  1. Please consider to revise the tittle, abstract, and conclusion to include CGA-rich extracts
  2. Figure 1: please add the atom/carbon number in each molecule figure. Readers can see the different between 5-CQA and 3-CQA/4-CQA; 3-FQA and 4-FQA from the atom/carbon number in the molecule figures. Please put the higher resolution figure for better clarity.
  3. LD50 -> please revise the 50 to subscript style.

Author Response

The authors have revised the manuscript based on the comments from the reviewer. There are only several minor changes needed:

We sincerely appreciate your comprehensive feedback on my manuscript again. I will address each of your suggestions individually in my revisions. Additionally, the alterations according to the comments in revised manuscript have been highlighted or deleted and that the revision has been enclosed in the submission system.

Comment 1: Please consider to revise the tittle, abstract, and conclusion to include CGA-rich extracts

Response 1: Thank you for your valuable feedback. While this review acknowledges studies that have utilized CGA-rich extracts (e.g., from coffee beans) to investigate the pharmacological effects of chlorogenic acid, their primary objective remains elucidating the pharmacological properties of CGA itself. It is worth emphasizing that CGA is a natural compound. As detailed in Section 2 of this article, various extraction methods have been employed to isolate CGA, and the majority of studies concerning its bioactivity have used highly purified forms of the compound. The focus of our review is to highlight the research value of CGA as a natural product—rather than as some extracts. We have also refined the wording in the abstract to better reflect this emphasis. It can be found in the line 4, 5 of the abstract.

Comment 2: Figure 1: please add the atom/carbon number in each molecule figure. Readers can see the different between 5-CQA and 3-CQA/4-CQA; 3-FQA and 4-FQA from the atom/carbon number in the molecule figures. Please put the higher resolution figure for better clarity.

Response 2: We sincerely appreciate your suggestion. We have further modified Figure 1 (significant carbon number) to allow for clearer differentiation of CGAs.

Comment 3: LD50 -> please revise the 50 to subscript style.

Response 3: Thank you for point this out, and we have modified. You can find it in the line16 of the first paragraph of section 5.